# RNA-directed remodeling of the HIV-1 protein Rev orchestrates assembly of the Rev–Rev response element complex

**Bhargavi Jayaraman, David C Crosby, Christina Homer, Isabel Ribeiro, David Mavor, Alan D Frankel\***

Department of Biochemistry and Biophysics, University of California, San Francisco, San Francisco, United States

**Abstract** The HIV-1 protein Rev controls a critical step in viral replication by mediating the nuclear export of unspliced and singly-spliced viral mRNAs. Multiple Rev subunits assemble on the Rev Response Element (RRE), a structured region present in these RNAs, and direct their export through the Crm1 pathway. Rev-RRE assembly occurs via several Rev oligomerization and RNA-binding steps, but how these steps are coordinated to form an export–competent complex is unclear. Here, we report the first crystal structure of a Rev dimer-RRE complex, revealing a dramatic rearrangement of the Rev-dimer upon RRE binding through re-packing of its hydrophobic protein–protein interface. Rev-RNA recognition relies on sequence-specific contacts at the well-characterized IIB site and local RNA architecture at the second site. The structure supports a model in which the RRE utilizes the inherent plasticity of Rev subunit interfaces to guide the formation of a functional complex.

## Introduction

Retroviruses such as HIV have small genomes and utilize multiple strategies such as overlapping reading frames and alternative splicing, to encode their repertoire of proteins. The ~9 kb HIV-1 genome codes for 15 proteins expressed from a set of unspliced and singly-spliced mRNAs in addition to the fully spliced messages (*Frankel and Young, 1998*). These unspliced and singly-spliced species encode the viral structural and accessory proteins and genomic RNA needed to assemble new virions, but because they contain introns are typically retained in the nucleus for splicing. To export these RNAs to the cytoplasm, the viral protein Rev (*Figure 1A*), expressed from a fully spliced message, translocates into the nucleus, forms an oligomeric complex on a ~350 nucleotide, highly structured intronic RNA element, the Rev Response Element (RRE) (*Figure 1B*), and directs their export through the Crm1 nuclear export pathway (*Fornerod et al., 1997*; *Pollard and Malim, 1998*).

Rev cooperatively assembles on the RNA using its oligomerization and RNA-binding domains to form a Rev hexamer on a shorter, but functional ~240 nucleotide RRE (*Malim and Cullen, 1991*; *Zapp et al., 1991*; *Daugherty et al., 2008*, *2010a*). This assembled RNP then presents Leu-rich nuclear export sequence (NES) peptides (*Figure 1A*) to interact with the host export receptor, Crm1, targeting these RRE-containing mRNAs for export (*Fornerod et al., 1997*). In this paper we address the structural principles that govern assembly of the Rev oligomer onto the RNA and its consequences for Rev function.

The binding of Rev to the RRE is nucleated at the stem IIB hairpin, where the α-helical arginine-rich motif (ARM) of Rev (*Figure 1A,B*) binds in the major groove of the RNA and makes an extensive set of sequence-specific and electrostatic contacts (*Malim et al., 1990*; *Cook et al., 1991*; *Heaphy et al., 1991*; *Kjems et al., 1991*; *Battiste et al., 1996*). The oligomeric complex proceeds to form by the sequential addition of Rev subunits (*Pond et al., 2009*). Crystal structures of Rev oligomerization surfaces depict a central hydrophobic core that mediates interactions between Rev subunits and positions the ARMs

**\*For correspondence:** frankel@cgl.ucsf.edu

**Competing interests:** The authors declare that no competing interests exist.

**eLife digest** To be able to multiply, viruses have to first infect a host cell and then hijack the host's molecular machinery to make viral proteins. One stage of this process takes place in the nucleus of the host cell and involves the viral DNA being transcribed to make RNA molecules. These RNA molecules must then be exported from the nucleus to the cytoplasm, where the viral proteins are made.

In the case of HIV-1, a protein called Rev has an important role in the export process. The Rev protein, which is supplied by the virus, binds to a region on the viral RNA molecules called the Rev Response Element. The Rev protein then binds to a group of host proteins called the Crm1 export complex to send the viral RNA molecules to the cytoplasm.

Jayaraman et al. now provide the first in-depth 3D structure of two Rev molecules bound to a fragment of the Rev Response Element. The Rev molecules change shape when they bind to the element, and specific regions of the element were found to be important for this. The experiments suggest that the Rev Response Element directs the positioning of the Rev proteins on itself to match the shape needed to bind to Crm1 export complex. In parallel work from the same laboratory, Booth et al. have produced a 3D structure of the whole complex.

Both structures shed new light on how the HIV-1 virus is able to multiply in its host, which may aid future efforts to develop new treatments for the disease.

on one side of the Rev oligomer to bind RNA and the NESs on the other side to recruit Crm1 (*DiMattia et al., 2010*; *Daugherty et al., 2010b*), highlighting the modular nature of the Rev domains.

In addition to the modular architecture of Rev, the RNA scaffold also plays a key role in defining the organization of the complex. Biochemical studies have shown that the RRE controls the oligomeric state of Rev (*Daugherty et al., 2010a*), and a recent SAXS structure shows that the RRE adopts an 'A' shaped architecture poised to recruit Rev (*Fang et al., 2013*). It remains unknown how the RNA structure is matched to the arrangement of the Rev oligomer to form the export–active complex.

One particularly fascinating question is how the individual subunits of Rev recognize the RRE. An NMR structure of a single α-helical ARM bound to the IIB RNA hairpin has been known for quite some time (*Battiste et al., 1996*), and a second binding site in stem IA was identified that utilizes a different surface of the helical ARM from another subunit to recognize the RNA (*Daugherty et al., 2008*). Other binding sites in the RRE may use yet other recognition strategies that have not been defined. As a step towards understanding how individual binding sites are organized on the RRE, how their arrangement relates to the architecture of the Rev oligomer, and how the different sites are recognized by Rev, we solved the structure of a Rev dimer bound to an RRE fragment (IIB40) containing two Rev-binding sites (*Daugherty et al., 2008*) (*Figure 1C*). This first high-resolution structure of a Rev-RRE complex (*Figure 1D*) uncovers yet another RNA recognition mode for Rev and shows how the remarkable plasticity of the Rev dimer allows it to adapt to the RNA framework.

## Results

### Rev dimer-RRE structure

To better understand how Rev assembles an oligomeric complex on the RRE, we sought to determine the crystal structure of a Rev-RNA complex. Rev has a strong tendency to aggregate and fold incorrectly when not bound to RNA (*Daugherty et al., 2010b*). We circumvented these problems by making RNA complexes with Rev 1–70 harboring the oligomerization-disrupting mutations, Leu12Ser and Leu60Arg (*Jain and Belasco, 2001*), which forms a stable Rev dimer (*Daugherty et al., 2010b*), and a Glu47Ala surface-entropy reducing mutation (*Goldschmidt et al., 2007*), which aided in crystallization (*Figure 1A*, *Figure 1—figure supplement 1*).

For the RNA, we utilized variants of RRE IIB40, which has the high-affinity IIB site positioned adjacent to a second site (referred to as the junction site) that mimics the stem II junction (*Daugherty et al., 2008*) (*Figure 1C*, *Figure 1—figure supplement 1*), and variants of a 68-nucleotide fragment comprising stems IIA, IIB and IIC. Previous biochemical studies have shown that Rev assembly initiates at stem IIB and proceeds through stem II along the length of stem I with the second Rev molecule

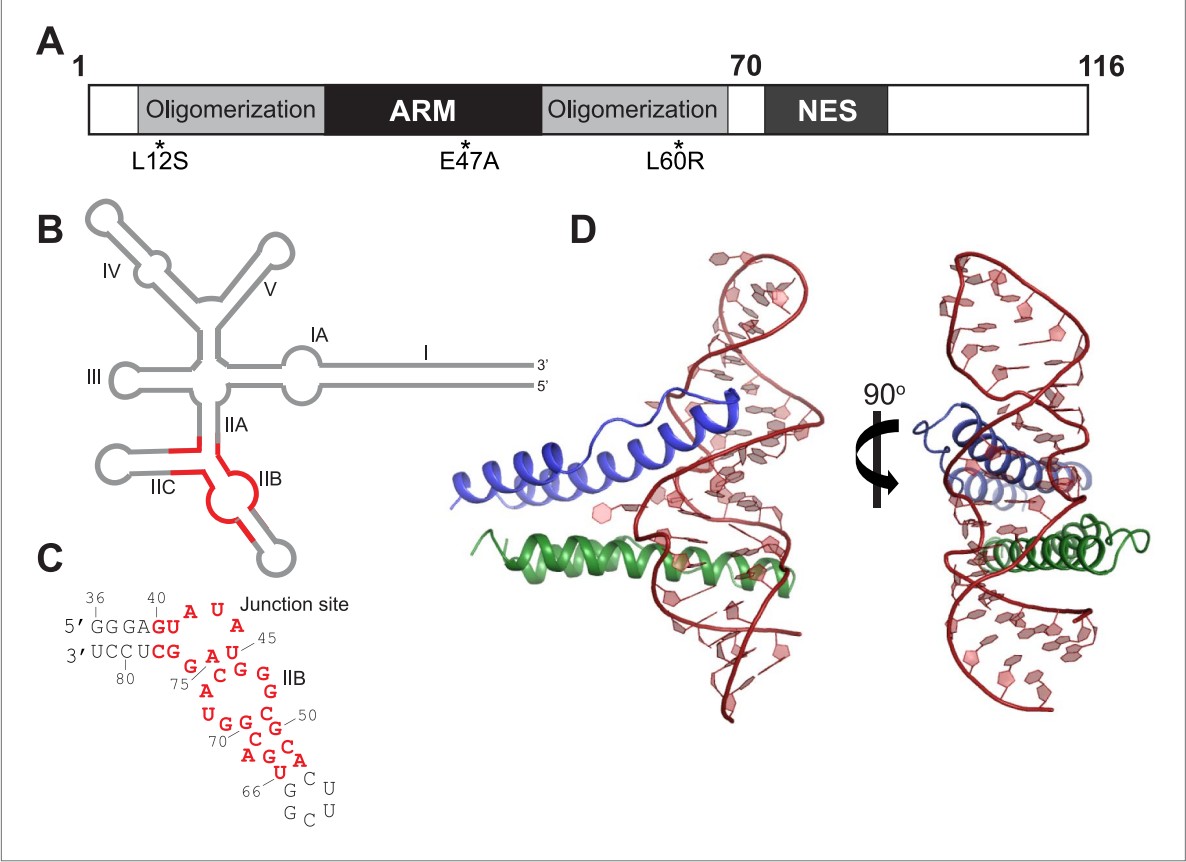

**Figure 1**. Overall organization and structure of the Rev dimer-RRE complex. (**A**) Domain organization of full-length Rev. The protein used to co-crystallize the complex contains residues 1–70 and contains the higher-order oligomerization disrupting mutations, L12S and L60R, and the surface-entropy reducing mutation, E47A. (**B**) Secondary structure of the RRE (**Watts et al., 2009**; **Bai et al., 2014**) with the region used for the co–crystal structure shown in red. (**C**) Sequence of IIB40 RNA with the region corresponding to the RRE shown in red. The bases are numbered according to older studies (**Battiste et al., 1996**) for consistency. The RNA contains a terminal UUCG tetra-loop to enhance stability and favor crystallization. See also **Figure 1—figure supplement 1** (**D**) Overall arrangement of the Rev dimer-RNA complex: The RNA (red) is held between the first Rev molecule (blue) bound at IIB and the second Rev molecule (green) bound at the junction site. The N-terminal 11 residues and C-terminal 7 residues of Rev 1–70 are not visible in the structure. See also **Figure 1—figure supplement 2, 3**.

The following figure supplements are available for figure 1:

**Figure supplement 1**. Rev and RRE constructs.

**Figure supplement 2**. Representative electron density maps.

**Figure supplement 3**. Comparison of Rev monomers.

binding at the stem II junction (**Mann et al., 1994**; **Charpentier et al., 1997**). The junction site was designed based on those studies (**Daugherty et al., 2008**) and also on the observation that Rev binding melts a A-U base-pair in stem IIA at the junction (**Charpentier et al., 1997**), a finding recently confirmed by SHAPE-seq experiments (**Bai et al., 2014**). Gel shift assays show that Rev binds to RRE stem II and to IIB40 with comparable affinity and cooperativity (**Figure 1—figure supplement 1**), suggesting that IIB40 recapitulates Rev assembly at the entire stem II junction.

We attempted co-crystallizations using ~35–40 RNA variants with altered stem lengths and tetra-loop sequences and obtained well-diffracting crystals with only a single Rev dimer-IIB40 complex. We solved its structure using single-wavelength anomalous dispersion to 5.5 Å to generate models for molecular replacement at 3.2 Å, resulting in the first high-resolution view of a Rev-RRE complex (**Figure 1D**, **Figure 1—figure supplement 2**, **Table 1**).

**Table 1.** Diffraction data collection and refinement statistics

| | Native | Tungsten derivative |
|---|---|---|
| Data Collection | | |
| Space group | P $4_1$ 3 2 | P $4_1$ $2_1$ 2 |
| Cell Dimensions | | |
| a, b, c (Å) | 165.3, 165.3, 165.3 | 147.22, 147.22, 199.44 |
| α, β, γ (°) | 90, 90, 90 | 90, 90, 90 |
| Resolution (Å) | 45.85 − 3.2 (3.314 − 3.2)* | 49.34 − 5.55 (5.75 − 5.55) |
| Redundancy | 25.9 (15.9) | 15.2 (13.6) |
| Completeness (%) | 99.72 (98.36) | 99.6 (96.5) |
| $I/\sigma I$ | 21.56 (1.74) | 13 (1.8) |
| R-meas | 0.138 (2) | 0.151 (1.52) |
| Refinement | | |
| Resolution (Å) | 45.85 − 3.2 (3.31 − 3.2) | |
| No. reflections | 342771 (20153) | |
| $R_{work}$/$R_{free}$ | 19.3 (27.7)/21.1 (30.7) | |
| No. atoms | | |
| RNA | 849 | |
| Protein | 862 | |
| Ligand/Ion | 5 | |
| Water | 1 | |
| B-factors | | |
| RNA | 113.9 | |
| Protein | 115.3 | |
| Ion | 153.8 | |
| Water | 112.7 | |
| RMS deviations | | |
| Bond lengths (Å) | 0.002 | |
| Bond angles (°) | 0.39 | |

*Statistics for the highest-resolution shell are shown in parentheses.

The Rev dimer-RRE structure depicts essential early steps in the assembly of the larger oligomeric complex. The overall architecture of the complex shows the RNA held between two Rev molecules arranged slightly asymmetrically in a V-shaped topology, with the two Rev ARMs buried in distorted major grooves positioned on opposite sides of the RNA (*Figure 1D*). The structures of the individual Rev monomers are nearly identical to each other and to the RNA-free form (*Figure 1—figure supplement 3*). Comparison of the structures of the free and RNA-bound Rev dimers, described below, illustrates how the homo-oligomer adapts to bind RNA cooperatively.

## The Rev dimer rearranges upon RNA binding

The Rev dimer in its RNA-free state also displays a V-shaped topology (*DiMattia et al., 2010*; *Daugherty et al., 2010b*) but its conformation changes dramatically upon RNA binding, with the two RNA-binding helices coming much closer together in the RNA-bound complex (*Figure 2A*). The Rev subunit bound to the junction site forms an extensive surface with the RNA that causes the dimer interface to pivot around a single residue, Ile55 (*Figure 2—figure supplement 1*), known to be critical for dimer integrity and function (*Jain and Belasco, 2001*; *Edgcomb et al., 2008*). This altered conformer results in complete repacking of hydrophobic residues at the dimer interface (*Figure 2B*), formed by residues Leu18 and Leu22 from the first helix and Ile55 and Ile59 from the second helix. Phe21, which formed a significant part of the RNA-free interface, is largely excluded.

Interestingly, the bound conformation buries less surface area at the dimer interface than the free form (1000 Å$^2$ vs 1500 Å$^2$) and also shows looser hydrophobic packing (the two Rev subunits are ~3.2 Å further apart) (*Figure 2—figure supplement 1*), suggesting that this bound dimer is energetically less favorable but is compensated by the energy of RNA binding of the second Rev subunit. The weaker interface may be further compensated by hydrogen bonding between the well-conserved Gln51 residues symmetrically located between the two subunits and observed only in the bound form (*Figure 2C*). Thus, mutation of Gln51 to alanine results in accumulation of Rev monomer-RNA complexes and ~30-fold reduced affinity (*Figure 2D*). In the context of the full-length hexameric Rev-RRE complex, the effect of the Gln51Ala mutation was modest but reproducible, showing a twofold reduction in affinity in gel shift assays using a vast excess of protein over RNA (*Figure 2—figure supplement 1*) and reduced accumulation of dimer species in assays with stoichiometric amounts of Rev and RRE (*Figure 2E,F*). The modest effect of the mutation in the full-length context suggests that the Gln51 interaction may be less crucial during later stages of Rev-RRE assembly and the energetic contributions to cooperative binding may shift to other interactions, possibly coupled to further rearrangements of the Rev subunit interfaces. Alternatively, there may be multiple pathways towards cooperative assembly such that if one pathway is blocked, another may be used.

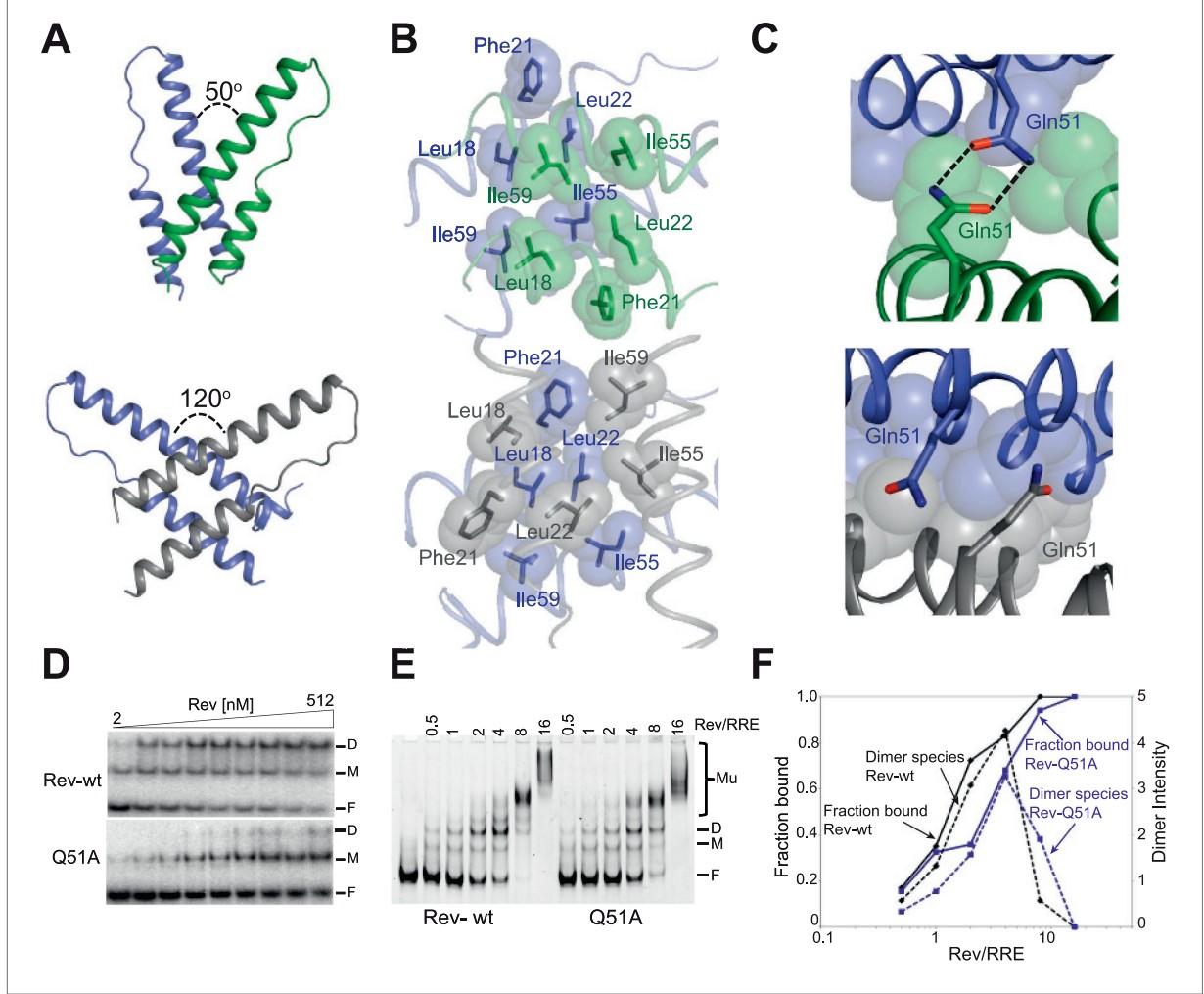

**Figure 2**. Reorganization of the Rev dimer interface upon RNA binding. (**A**) Rev dimer crossing angles differ significantly between RNA-bound (top) and RNA-free (bottom) states (**B**) Packing of hydrophobic residues at the dimer interface in the RNA-bound (top) and RNA-free (bottom) states. Phe21 is largely excluded from the interface upon RNA binding and, in general, the interface is more loosely packed (*Figure 2—figure supplement 1*). (**C**) Gln51 hydrogen bonds across the dimer interface in the RNA-bound conformation (top) but is unable to interact in the free state (bottom). (**D**) Gel shift assays with ³²P-labeled IIB40 RNA and Rev 1–70 dimer show that a monomeric Rev-RNA complex accumulates when Gln51 is mutated to Ala and the dimer affinity is reduced ~30-fold (*Figure 4—figure supplement 1*). F corresponds to free RNA, M to the Rev monomer-RNA complex, and D to the Rev dimer-RNA complex. (**E**) Gel shift assays with the 234-nt RRE and full-length Rev, visualized using SYBR Green II staining, show reduced accumulation of dimer species and a modest loss of binding affinity with the Gln51Ala mutant (*Figure 2—figure supplement 1*), quantified in panel (**F**). Mu corresponds to Rev multimer-RNA complexes.

The following figure supplement is available for figure 2:

**Figure supplement 1**. Rev interactions in the free and RNA-bound structures.

## Diversity of RNA recognition by the Rev ARM

The pliability of the Rev dimer interface contributes to RNA-binding specificity by allowing the Rev subunits to orient properly to multiple binding sites arrayed on the RRE, with binding at the primary IIB RNA site nucleating assembly. Comparison of our Rev dimer-IIB40 structure to the NMR structure of a Rev ARM peptide-IIB complex (*Battiste et al., 1996*) shows that both the Rev arginine-rich-helix and IIB RNA site are remarkably similar in both the structures (*Figure 3A*), although with subtle changes to some Rev-IIB contacts (*Figure 3B*, *Figure 3—figure supplement 1*). For example, in the NMR structure, Asn40 hydrogen bonds to a G47-A73 base pair in IIB RNA, but is in a different plane and contacts G47 and G71 from adjacent base pairs in the dimer-RNA complex (*Figure 3B*). Interestingly, the

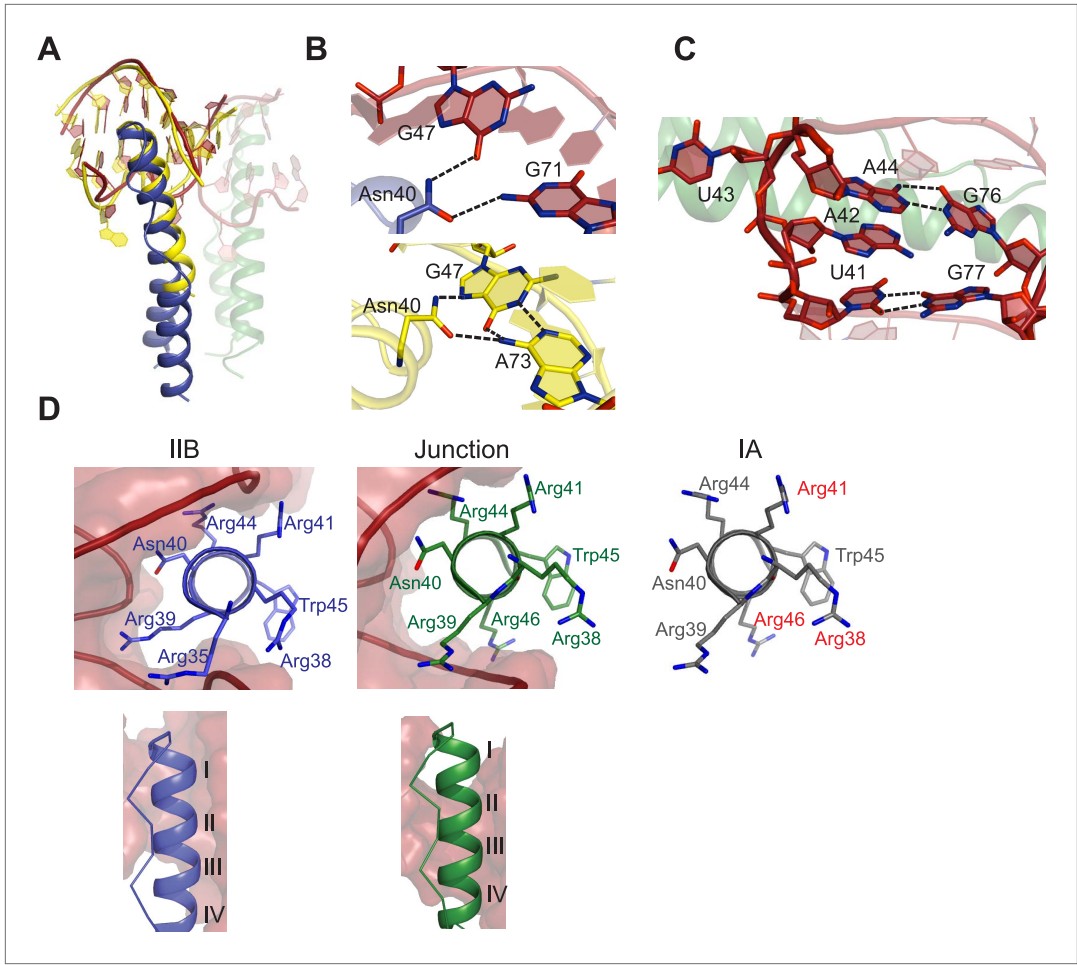

**Figure 3**. Features of Rev-RNA recognition. (**A**) The ARM peptide-IIB NMR structure (yellow) (***Battiste et al., 1996***) and the Rev dimer-IIB40 structure are nearly superimposable (backbone RMSD = 1.27 Å). (**B**) Contacts made by Asn40 from the first Rev subunit at the IIB RNA site are different in the Rev dimer-IIB40 (top panel) and ARM peptide-IIB complexes (bottom panel). See also ***Figure 3—figure supplement 1***. (**C**) RNA structure at the second junction site shows a G-A base-pair and A42-U43 bulge that widens the major groove to accommodate the RNA-binding helix of the second Rev subunit. See also ***Figure 3—figure supplement 2, 3***. (**D**) Rev- RNA recognition at the three known binding sites in the RRE, shown in the three top panels as views down the helical axis of the Rev ARM at the IIB site, junction site, and stem IA. The binding residues for stem IA are inferred from alanine mutants (in red) (***Daugherty et al., 2008***). The two bottom panels show side views from the co–crystal structure indicating that turns 1–3 of the ARM helix contact IIB while turns 2–4 contact the junction site.

The following figure supplements are available for figure 3:

**Figure supplement 1**. Schematic representation of Rev-RNA contacts at IIB and junction sites.

**Figure supplement 2**. Rev-RNA recognition at the junction site.

**Figure supplement 3**. Model for RNA-directed cooperative assembly.

Asn40-IIB interaction observed in the crystal structure is similar to that seen in an NMR structure of a Rev-ARM peptide-aptamer complex (***Ye et al., 1996***), suggesting that optimal Rev-RNA contacts can be selected from several favorable conformers. The subtle differences between the NMR and crystal structures likely reflect a lack of precision of the early NMR data, with no violations observed to the current structure. The similarity in Rev-IIB recognition between the NMR structure, which depicts the first step in Rev-RRE assembly, and the crystal structure, which captures the progression into

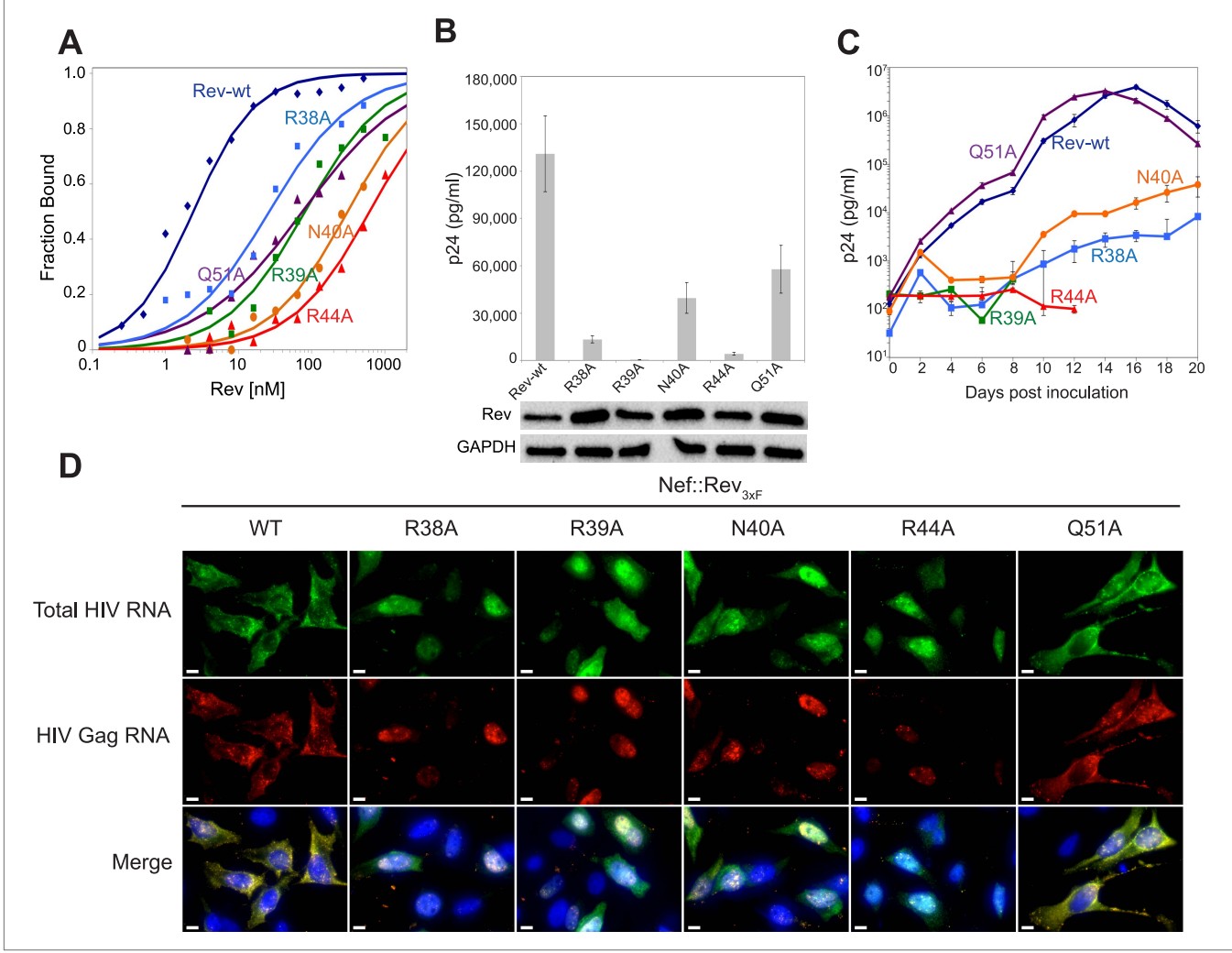

**Figure 4**. Role of RNA-binding and dimer interface residues in Rev function. (**A**) Representative binding curves with Rev 1–70 dimer mutations at positions observed to contact the RNA, calculated from gel shift assays using $^{32}$P-labeled IIB40. Apparent dissociation constants, $K_d$ and Hill coefficient (n) are reported in *Figure 4—figure supplement 1*. (**B**) p24 production from a Rev-dependent, transiently transfected pCMV GagPol-RRE reporter assay complemented in trans with full length Rev mutants, quantified by ELISA and normalized to both Rev and GAPDH expression levels (representative western blots below show that the mutations do not affect Rev steady state expression levels or stability). Data points are mean ± s.d. of biological triplicates. (**C**) Viral replication kinetics for HIV-1$_{NL4-3}$ containing Rev mutants, monitored by p24 release into the culture supernatant. Data points are mean ± s.d. of biological triplicates. See also *Figure 4—figure supplement 2* (**D**) FISH of total HIV RNA or unspliced Gag RNA in the presence of the indicated Rev mutation. Nuclei were stained using DAPI (blue). Nef::Rev$_{3xF}$ denotes infection with virus containing c-terminally 3× flag-tagged Rev in the *nef* locus. See also *Figure 4—figure supplement 3*.

The following figure supplements are available for figure 4:

**Figure supplement 1**. Gel-shift assays with the indicated Rev mutants.

**Figure supplement 2**. Design of virus constructs.

**Figure supplement 3**. RNA FISH and protein immunofluorescence studies.

cooperative assembly, reiterates the importance of preserving this highly sequence-specific interface to support a specific and productive assembly.

The junction site in the RNA forms a structural scaffold that helps orient the Rev subunits. It is formed by an A44-G76 base-pair, identified earlier by in vitro selection studies (*Bartel et al., 1991*), and a A42-U43 bulge that widens an otherwise narrow A-form major groove to accommodate the

α-helix of the second Rev subunit (*Figure 3C*, *Figure 3—figure supplement 2*). Both A42 and U43 are unpaired and A42 is co-axially stacked on stem IIB, consistent with biochemical mapping experiments showing increased accessibility of these nucleotides upon Rev binding (*Charpentier et al., 1997*; *Bai et al., 2014*). Neither base is directly contacted by Rev, suggesting that their primary role is to create the junction site architecture. Consistently, deleting the bulge results in exclusively monomeric binding (*Figure 3—figure supplement 2*), and previous studies showed that simply introducing bulges with variable sequences juxtaposed to IIB is sufficient to mediate co-operative Rev binding (*Zemmel et al., 1996*). Most contacts at the junction site are to the phosphate backbone with the exception of Arg 43 and Arg44 (*Figure 3—figure supplement 1*). Notably, Arg44 from both Rev subunits contacts the two binding sites on the RNA, in one case using water-mediated interactions not previously observed for RRE recognition (*Figure 3—figure supplement 1, 2*).

The junction site in the full-length RRE is formed by three stems (IIA, IIB, and IIC), with only IIB and IIC present in the crystallized IIB40 RNA. Stem IIA likely forms an A-form helix that merges into the junction to determine the relative placement of the remaining RRE. The second Rev subunit is positioned in the crystal structure to use its higher order oligomerization surface to recruit a third Rev subunit with its ARM oriented towards stem IIA for sequential Rev assembly. Presumably, the overall 'A'-shaped RRE topology (*Fang et al., 2013*) and arrangement of binding sites dictates the positioning of newly recruited subunits (*Figure 3—figure supplement 3*). The specific model for Rev-RRE assembly proposed by *Fang et al. (2013)* portrays an initial Rev dimer bridging the RRE IIB and IA sites, but our structure and other biochemical studies of Rev-RRE assembly (*Charpentier et al., 1997*; *Bai et al., 2014*) suggest that the first Rev dimer binds to stem II before additional subunits are added. Rev dimers formed later during assembly are likely positioned to bridge the two arms of the 'A'-shaped RRE, presumably involving the IA site.

The diversity of RNA recognition by the Rev ARM is quite striking, now with examples of how Rev binds to three sites in the RRE, and yet other binding modes observed with in vitro selected nucleic acids (*Xu and Ellington, 1996*; *Ye et al., 1999*; *Bayer et al., 2005*). The current crystal structure shows that each of the two ARMs of the Rev dimer deeply insert into a distorted RNA major groove, each burying ~1500 Å$^2$ of surface area, and that the same face of the ARM is used to contact the RNA in both cases (*Figure 3D*). However their helical registers differ and recognition of IIB is highly sequence-specific compared to the junction site. Furthermore, at another characterized Rev-binding site in the RRE, stem IA, yet a different binding mode seems to be utilized, with mutational studies indicating a different face of the ARM used to bind the RNA (*Figure 3D*) (*Daugherty et al., 2008*).

The Rev-RNA interface at the junction site illustrates that protein contacts at individual RRE sites can be largely sequence-nonspecific if the ARM is optimally presented to the RNA, especially from the context of additional bound Rev subunits. While binding to some sites such as stem IIB or IA can show substantial specificity even in their isolated contexts, this does not seem to be the case for the remaining Rev binding sites (*Daugherty et al., 2008*). Binding at these sites might require a more extensive RRE framework and neighboring Rev subunits and may differ from the configuration seen at the junction site, potentially adding further diversity to the recognition modes utilized by the ARM.

## Contributions of RNA-binding and dimer interface residues to Rev function

To test the functional implications of the observed interactions in the crystal structure, we monitored the effects of key mutations in Rev on RRE binding, viral RNA export, and viral replication. A previous study identified Asn40 as the most critical residue for IIB recognition, followed by Arg44, Arg38 and Arg39 (*Tan et al., 1993*). In the dimer-RNA complex however, alanine substitution of Arg44, which contacts both RNA sites, was most deleterious (250-fold reduced affinity) followed by Asn40 (100-fold), Arg39 (30–50-fold), and Arg38 (10-fold) (*Figure 4A*, *Figure 4—figure supplement 1*). These mutations, as well as mutation of Gln51 which hydrogen bonds across the dimer interface, were all defective in viral RNA export and translation in Rev-RRE-dependent Gag-Pol reporter assays (*Figure 4B*). When engineered into HIV-1, all mutations, except Gln51, also showed attenuated viral replication in tissue culture (*Figure 4C*, *Figure 4—figure supplement 2*). As noted above, Gln51 is important at the level of dimer formation but may be less critical in the full Rev-RRE context. RNA fluorescence in situ hybridization (FISH) of infected cells indicated that these Rev mutations caused retention of unspliced HIV-1 RNAs in the nucleus (*Figure 4D*), correlating with reduced Gag protein production

(*Figure 4—figure supplement 3*). Thus, key interactions observed in the Rev-RNA crystal structure have important functional roles in HIV RNA export and viral replication.

## Discussion

Previous crystal structures of Rev dimers and the NMR structure of a Rev ARM-IIB complex defined key building blocks of Rev-RRE assembly, but it has been unclear how they are employed to guide cooperative RNA binding of the oligomer to assemble a fully functional complex. Our structure of the Rev-RRE complex illustrates a direct physical coupling between RNA binding and protein oligomerization, where RNA binding drastically alters the conformational state of the Rev dimer. The architecture of the RRE and positioning of individual Rev-binding sites configures the predominantly hydrophobic protein–protein interfaces to form a cooperative complex. This is reminiscent of the DNA-induced heterodimerization of nuclear receptors, where DNA sequence and spacing between protein-binding half-sites configures the dimer interface and dictates the choice of dimerization partners (*Meijsing et al., 2009*; *Rastinejad et al., 2013*).

The use of non-polar residues at protein–protein interfaces has been postulated to be an attractive choice for evolutionary change as it can readily accommodate structural perturbations (*Lesk and Chothia, 1980*). Structural studies of the HIV-1 capsid protein clearly illustrate the role of plasticity at interfaces to assemble complex structures such as fullerene cones (*Byeon et al., 2009*; *Pornillos et al., 2009*, *2011*). The simple modular structure of Rev, including its pliable protein–protein interface and diverse RNA-binding domain, highlights how a virus with limited coding capacity can build a large, asymmetric RNP using a small, homo-oligomeric protein to achieve remarkable structural and functional complexity.

Our structure expands the repertoire of known Rev dimer conformers to three, each with different crossing angles between the RNA-binding helices. Given the hydrophobic nature of the interfaces, other conformers are possible, especially when placed in the context of host–protein complexes. These conformers can be combined in multiple arrangements to generate oligomers with diverse architectures (*Figure 5A*). Since Rev likely interacts with other host proteins to transport and package viral RNAs during the virus life cycle (*Jager et al., 2012*; *Naji et al., 2012*), different quaternary structures of Rev may generate conformational states adapted to different host complexes at various stages of viral replication. This is highly reminiscent of the Ebola virus protein VP40, whose dimeric, hexameric, and octameric forms each have different roles in the viral life cycle (*Bornholdt et al., 2013*), again highlighting the value of adaptable homo-oligomers in small RNA viruses. Recent crystal structures of HIV Tat (*Tahirov et al., 2010*) and Vif (*Guo et al., 2014*) with host protein complexes illustrate the importance of scaffolding structures to enable viral proteins to adopt ordered, functional states. It is remarkable that for Rev, it is not the folding of a single polypeptide chain that is dictated by its scaffolding partner(s) but rather its oligomeric architecture.

The function of Rev in RNA export is well defined, and the arrangement of Crm1 export complexes provides one clear example in which RNA organizes the Rev oligomer into a functionally important state. The 'jellyfish' model depicts Rev hexamers to be associated with the RNA on one side of the protein and NESs that recruit Crm1 export receptors on the other side (*Daugherty et al., 2010b*). Recent structural studies show that the Rev-RRE complex binds a Crm1 dimer utilizing two NES-binding pockets positioned for Rev recognition (*Booth et al., 2014*). This study also shows that the RRE plays an important role in assembly of the export complex by enhancing the affinity of the Rev oligomer for Crm1, probably by using the RNA scaffold to organize the Rev subunits and increase the local concentration of NES peptides for Crm1 binding. Additionally, SAXS studies of the RRE (*Fang et al., 2013*) and SHAPE-seq analysis of Rev-RRE assembly (*Bai et al., 2014*) reveal that the RNA is compact and pre-ordered to bind Rev. Consequently, it may be inferred that changes to the RRE scaffold can affect the overall arrangement of Rev subunits and redistribute the spatial availability of NESs for Crm1 recruitment and nuclear export (*Figure 5B*). Multiple studies support this hypothesis: (1) Resistance to a dominant negative mutant of Rev with a defective NES arose through two mutations in the RRE that changed the RNA structure but not Rev multimerization (*Legiewicz et al., 2008*), consistent with an RNA-guided reorganization of Rev oligomers. (2) Conversely, Rev complexes formed with simple repeats of the IIB hairpin displayed high RNA-binding affinity but did not recapitulate full export activity (*Symensma et al., 1999*), indicating that the specific RRE architecture properly arranges the Rev oligomer for function. (3) A study of cognate Rev-RRE pairs from HIV-infected patients found that mutations in the RRE were responsible for changes in activity as the pairs evolved during the course of infection (*Sloan et al., 2013*), suggesting that small changes in RRE

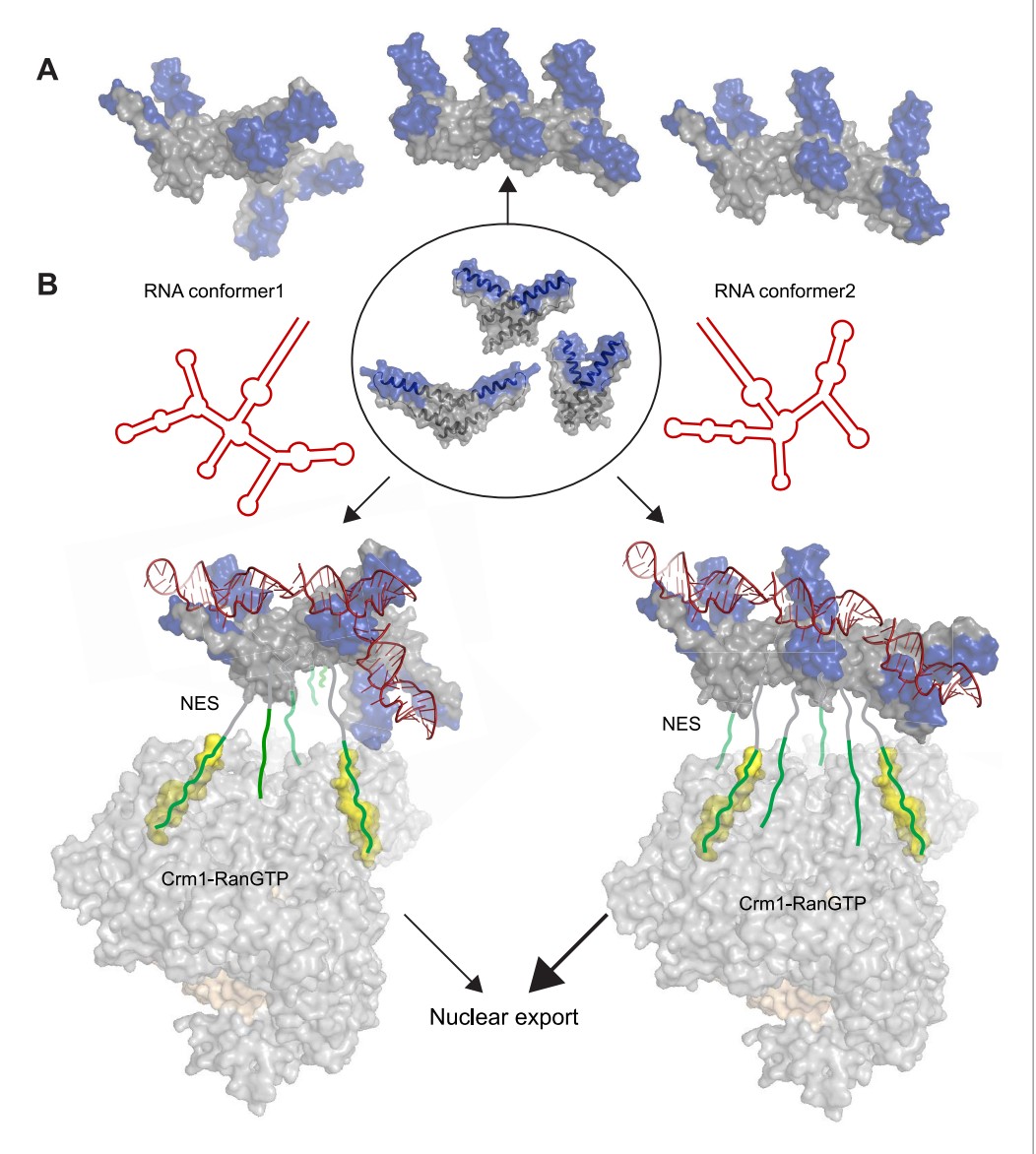

**Figure 5**. Potential diversity of Rev oligomeric structures and functional implications. Three types of Rev–Rev interactions observed by crystallography, Rev dimers in the RNA-free (*Daugherty et al., 2010b*) or RNA-bound states (the current structure) and a Rev dimer using the higher-order oligomer interface (*DiMattia et al., 2010*) are shown within the circle with Rev in grey and the Rev-ARMs in blue. (**A**) The higher-order oligomer interface was used to combine Rev dimers in the RNA-free state or RNA-bound state in various arrangements to generate examples of hexamers with different architectures. (**B**) Models illustrating how changes to the RRE structure (red) can alter the architecture of Rev oligomer. Such changes can alter the Rev-RRE 'jellyfish' architecture and spatial distribution of NESs, potentially changing their local effective concentration and avidity for the Crm1 dimer (in grey, with RanGTP in light brown and NES-binding sites in yellow), thereby tuning nuclear export activity. Coordinates of the Crm1-RanGTP dimer are from *Booth et al., 2014*.

sequence can alter the architecture of the Rev-RRE complex to tune its functional output. Previous studies likened the RRE to a molecular rheostat, where it is sensitive to the intracellular Rev concentrations and export activity adjusts accordingly (*Mann et al., 1994*). It appears now that RRE can also perform as a structural/evolutionary rheostat, where changes to RRE structure during the course of infection can rearrange the Rev oligomer and tune the levels of nuclear export. It will be interesting to determine if interactions with other host proteins exploit the Rev-RRE oligomer in other ways.

# Materials and methods

## Protein expression and purification

We used a Rev 1–70 dimer construct harboring the higher-order oligomerization-disrupting mutations, Leu12Ser and Leu60Arg (*Daugherty et al., 2010b*), and a Glu47Ala surface-entropy reducing mutant (*Goldschmidt et al., 2007*) for initial crystallization screens. We note that position 47 in most HIV-1 Rev isolates already is alanine, unlike the Glu47 found in HIV strain HXB3. Rev protein was expressed in *Escherichia coli* BL21 (DE3) with an N-terminal hexa-histidine tag and GB1 solubility-enhancing domain and purified using Ni affinity chromatography essentially as described (*Daugherty et al., 2010a*). However, instead of using RNAses to eliminate contaminating endogenous *E. coli* RNA that co-purifies with Rev, we added NaCl to 2 M and urea to 1 M to the cleared lysate (*Marenchino et al., 2009*) and incubated for 1–2 hr at 4°C while binding in batch to the Ni-NTA resin. Washes and elutions were performed as described (*Daugherty et al., 2010a*). Fractions of pure protein were pooled, concentrated to ~0.2 mM and purified by size-exclusion using a Superdex 75 column equilibrated in 40 mM Tris pH 8.0, 0.2 M NaCl, 0.1 M $Na_2SO_4$. To remove the His-GB1 tag, protein fractions after size-exclusion were pooled and cleaved with Tev protease after adding ammonium sulfate to 0.4 M. The pure, tag-free Rev obtained was concentrated to 40–50 µM and stored at 4°C prior to complex formation with RNA.

## RNA synthesis

RNAs were synthesized by T7 in vitro run-off transcription from synthetic DNA templates as described (*Daugherty et al., 2008*). Purified and annealed RNAs were lyophilized and stored at −20°C.

## Crystallization

Lyophilized RNAs were resuspended in water at 0.4–0.5 mM. Protein and RNA were incubated together at 1.9:1 ratio at room temperature for 30 min, concentrated to 5–6 mg/ml using an Amicon ultrafree 4 3 kDa cut-off concentrator (EMD Millipore, Billerica, MA), exchanged into crystallization buffer (10 mM HEPES pH 6.5, 50 mM KCl, 2 mM $MgCl_2$) using zeba desalting columns (Thermo Fisher Scientific, Rockford, IL) and concentrated to 10–12 mg/ml. Initial screens were performed using a Mosquito robot with ProComplex and Nucleix suites (Qiagen Inc, Valencia, CA). After screening ~35–40 Rev dimer-RNA complexes, one complex comprising the Rev dimer Glu47Ala mutant and an RNA derived from IIB40, containing a UUCG tetraloop, formed cubic crystals in several conditions with PEG 400, PEG 4000 and PEG 8000. Optimized crystals were grown using hanging drop vapor diffusion by mixing the complex at 1:1 with reservoir solution containing 50 mM MES 6.0, 50 mM KCl 1–4% PEG 4000 (for native diffraction data collection) or 50 mM sodium cacodylate pH 6.0, 5 mM $MgCl_2$, 8–14% PEG 400 (for heavy atom soaking). Crystals appeared in 1–2 days and grew to their full size of 100–200 microns in 3–4 days. After 1 week, native crystals were transferred into a drop containing well solution supplemented with 5% PEG 4000. PEG 400 and PEG 4000 were added directly to the drop in small increments to a final concentration of 25% PEG 400 and 14% PEG 4000 after which the crystals were flash frozen in liquid nitrogen. Crystals grown in MES buffer failed to survive heavy atom soaks but crystals grown in 50 mM cacodylate pH 6.0, 5 mM $MgCl_2$, 8–14% PEG 400 were more tolerant to heavy atom soaks although they did not diffract beyond 4.5 Å. Crystals for heavy atom soaking were transferred to a fresh drop of well solution supplemented with 5% PEG 4000 and PEG 400–25% was added to the drop in small increments. For anomalous data collection, crystals were soaked in 7.1 mM ammonium tetrathio tungstate (Hampton Research, Aliso Viejo, CA, Heavy Atom Screen M2) for 15 min before flash freezing in liquid nitrogen.

## Diffraction data collection and structure determination

Data sets were collected at the Advanced Light Source beamline 8.3.1 at 100 K. Crystals belonged to either a tetragonal ($P4_12_12$) spacegroup or a cubic spacegroup ($P4_132$). Diffraction data was processed and scaled using XDS (*Kabsch, 2010*). Attempts with molecular replacement using the unbound Rev dimer structure or models of Rev-IIB RNA complexes as starting models with native diffraction datasets (collected at 1.116 Å) were unsuccessful. Using single wavelength anomalous dispersion (SAD) data to 5.5 Å on a W-soaked crystal (collected at 1.2146 Å), tungstate ions were located using ShelxD, phased and density-modified using ShelxE (*Sheldrick, 2010*). The crystal belonged to $P4_12_12$ space

group with three copies of the Rev dimer-RNA complex and one W site per asymmetric unit. Density for the Rev helices and RNA was very clear and Rev molecules from the Rev-dimer structure (*Daugherty et al., 2010b*) and an RNA fragment from IIB RNA NMR structure (*Battiste et al., 1996*) were fit into the protein and RNA densities, respectively, using rigid-body fit in Coot (*Emsley et al., 2010*). The resulting model was used for molecular replacement into the 3.2 Å native data in the cubic space group (P4₁32) in Phaser (*McCoy et al., 2007*). Initial solutions scored poorly but the appearance of positive density near the RNA fragment at the second Rev-binding site suggested that the solution was correct. Following iterative rounds of model building of the RNA using RCrane (*Keating and Pyle, 2010*) in Coot and refinement in Phenix (*Adams et al., 2010*), a second round of molecular replacement was carried out using the newly built RNA alone as the search model. This time, the solution was clear with positive density for both Rev subunits. Rev subunits were then built into the density with iterative model building and refinement. The final model had good stereochemistry and showed no Ramchandran outliers (*Chen et al., 2010*). While density for most water molecules or ions was not apparent at this resolution, we could place one water molecule bridging a Rev-RNA contact. Solvent-accessible surface areas were calculated using PISA (*Krissinel and Henrick, 2007*) and figures were generated using PyMOL (*Schrodinger, 2010*).

## Gel shift assays

Electrophoretic mobility shift assays were performed in gel shift buffer (10 mM HEPES pH 7.5, 0.1 M KCl, 1 mM MgCl₂, 0.5 mM EDTA, 2 mM DTT, 10% glycerol, 50 µg/ml yeast tRNA and 0.2 mg/ml BSA). Rev protein stock for RNA binding was prepared in gel shift buffer supplemented with fivefold molar excess of yeast tRNA. Rev was diluted in gel shift buffer and mixed with an equal volume of <25 pM ³²P-RNA. Reactions were incubated at room temperature for 15 min and loaded onto continuously running 6% (for 234 nucleotide RRE) or 10% polyacrylamide gels (0.5× TBE). Gels were run at room temperature for 60–90 min, dried and exposed to a phosphorimaging screen for >12 hr. Bands were quantified using ImageJ (*Schneider et al., 2012*) and data were fit using MS Excel.

For gel shift assays using RRE and detection with SYBR Green II staining (Invitrogen), gel shift buffer contained 10 mM HEPES pH 7.5, 0.3 M KCl, 1 mM MgCl₂, 0.5 mM EDTA, 2 mM DTT, 10% glycerol and 0.2 mg/ml BSA (*Fang et al., 2013*). Rev was serially diluted and mixed with an equal volume of RRE at 125 nM, incubated for 15 min and loaded onto continuously running 6% polyacrylamide gels (0.5× TBE). Gels were run at room temperature for 1–2 hr, stained with SYBR Green II and visualized under UV light. Bands were quantified using ImageJ (*Schneider et al., 2012*).

## Rev nuclear export activity reporter assay

The pCMV-GagPol-RRE reporter construct (*Srinivasakumar et al., 1997*) was modified to contain the HIV-1 SF2 RRE sequence (used in biochemical assays) and was complemented in trans with pcD-NA4TO-Rev-3xFlag (Invitrogen mammalian expression vector) in 293T cells by transient transfection. The optimal ratio of reporter to Rev expression plasmids was first determined to establish maximal signal under non-saturating conditions for the assay. Subsequently, 50,000 HEK 293T cells, each in 96-well microtiter plate wells, were transfected with 90 ng of reporter plasmid, 0.3 ng of Rev expression plasmid, 5 ng of an mCherry fluorochrome expression vector (to visualize transfection efficiency) and 4.7 ng of pBS carrier DNA. 48 hr after transfection, the culture supernatant was removed, the cells were lysed and intracellular p24 was quantified by ELISA. mCherry expression was consistent across all transfections at the time of cell lysis. The expression of Rev-3xFlag and GAPDH was monitored by western blot analyses using mouse α-Flag (Sigma-Aldrich Corp. St. Louis, MO) and mouse α-GAPDH (Abcam, Cambridge, MA) antibodies, respectively, followed by goat α-mouse IgG-HRP (Santa Cruz Biotechnology, Dallas, TX). HRP-bound antigens were then detected by chemi-luminescence and imaged using a BioRad ChemiDoc MP imager (Bio-Rad Laboratories, Inc. Hercules, CA). Expression levels for Rev mutants were within twofold of wild-type Rev, when normalized to GAPDH loading control, and were used to normalize the quantified p24 levels (*Figure 4B*).

## Viral replication assays
### Generation of proviral clones
The *rev* cDNA was relocated to the *nef* region (*nef* is dispensable for ex vivo HIV replication in SupT1 cells) (*Figure 4—figure supplement 2*) to facilitate mutagenesis of *rev* without affecting the overlapping *tat* or *env* reading frames.

## Cells and viruses

The infectious molecular clone, HIV$_{NL4-3}$, was acquired from the NIH AIDS Reagent Program. HEK 293T cells were co-transfected with proviral plasmids and mammalian expression vectors containing HIV$_{NL4-3}$ Rev and Tat at a ratio of 1:50 (wt/wt, mammalian expression vector:proviral plasmid) to promote robust production of viruses containing mutant Rev. Virus-laden supernatants were collected 48 hr post transfection, centrifuged at 500×$g$ for 5 min, 0.45 µM-filtered, and stored in aliquots at −80°C. Virus stocks were titered using ELISA to quantify p24 viral capsid protein. Mouse monoclonal and rabbit polyclonal α-p24 antibodies used in the ELISA were obtained from the NIH AIDS Reagent Program.

## Virus spreading assay

Sup-T1 cells (NIH AIDS Reagent Program) were cultured in RPMI media supplemented with 25 mM HEPES pH 7.4, 10% heat-inactivated fetal calf serum (Hyclone, Waltham MA), and 1% penicillin/streptomycin. 10$^6$ Sup-T1 T-cells were inoculated with 2.5 ng p24 in 250 µl media containing 1 µg/ml PEI and 8 µg/ml polybrene in 96-well flat bottom polystyrene cell culture microplates and then centrifuged for 2 hr at 1200×$g$ at 37°C. Each virus was assayed in biological triplicate. Input virus was then removed via washing cells twice with 250 µl phosphate-buffered saline (PBS). Cells were resuspended in 250 µl of media and incubated at 37°C, 5% CO$_2$. Under these centrifugal inoculation conditions, the multiplicity of inoculation (MOI) is approximately 0.05–0.025 as determined by acute infection of Sup-T1 T-cells with serial dilutions of HIV$_{NL4-3}$ expressing green fluorescent protein (GFP) followed by flow cytometry to quantify the GFP-expressing population. The infections were monitored every 48 hr following inoculation by immunofluorescence assay to detect cellular HIV antigen synthesis and by HIV p24 ELISA on the culture supernatant to quantify virus release. The infections were monitored until robust HIV cytopathic effect (cell necrosis and syncytia formation) was observed in the wild-type virus control.

## Protein subcellular localization and RNA fluorescence in situ hybridization (FISH) assays

HeLa cells (150,000) cultured on 12 mm poly-L-lysine coated glass coverslips were acutely infected with 100 ng p24 of HIV pseudotyped with VsV-G envelope glycoprotein and containing 3x-flag epitope-tagged Rev in the *nef* locus in DMEM media containing 8 µg/ml polybrene. These inoculation conditions equate to an effective MOI of 1.0–0.5 as determined previously via TCID$_{50}$ determination in HeLa cells.

24 hr post inoculation, the cells were washed twice with PBS, fixed in 4% paraformaldehyde/PBS for 10 min at RT, washed in 0.15 M glycine/PBS, permeabilized with 0.5% Triton X-100/PBS, and then blocked with 1% bovine serum albumin (BSA)/PBS/0.1% Tween-20 at 37°C for 1 hr. Antigens were then probed with a 1:250 dilution (vol/vol) of rabbit polyclonal α-HIV p24 (NIH AIDS Reagent Progam) and a 1:500 dilution of mouse α-flag (Sigma) in 1% BSA/PBS/0.1% Tween-20 at 37°C for 1 hr. The cells were washed twice with PBS/0.1% Tween-20 and then probed with a 1:250 dilution of goat α-rabbit IgG-rhodamine (MP Biomedicals, LLC, Santa Ana, CA), a 1:500 dilution of goat α-mouse IgG-FITC (MP Biomedicals) and 100 ng/ml DAPI nuclear stain. The cells were washed three times with PBS/0.1% Tween-20 and the coverslips were mounted to glass slides using Vectashield hardset mounting media for fluorescence. Cells were imaged using a 63×, 1.3 NA Leica objective and a Hamamatsu C4742-12 cooled CCD camera on a Leica DMI6000B stand. 25 z-stack images were taken using exposures normalized to the brightest sample, background-subtracted, and then merged to a single image using the brightest point projection method in the Volocity imaging software suite (Perkin Elmer, Waltham, MA).

For RNA subcellular localization by FISH, acutely infected HeLa cells were fixed with methanol for 10 min at RT and rehydrated in 2× SCC/10% formamide at 37°C for 10 min. They were then probed using Stellaris probe sets (Biosearch) targeting HIV-1 *gag* nt790–2292 conjugated to Quasar570 dye and HIV-1 *nef* nt8787–9407 conjugated to Quasar670 dye at 125 nM in 2× SCC/10% formamide/100 mg/ml dextran sulfate for 2 hr at 37°C. The total HIV-1 RNA probe set included 39 unique DNA oligonucleotides and the HIV-1 gag RNA probe set included 48 unique DNA oligonucleotides each computationally predicted to bind specifically to their target sequence (Biosearch Stellaris FISH Probe Designer web-based program). Following incubation, 2 ml of 2× SCC/10% formamide was added and incubated for 30 min at 37°C to dissociate nonspecifically bound probe. The wash was aspirated, and 1 ml of 2× SCC/10% formamide containing 100 ng DAPI nuclear stain was added and incubated at 37°C for 30 min. The cells were then washed twice with 2× SCC, coverslips mounted to glass slides,

and cells were imaged with brightness and contrast adjusted using the same linear parameters for all images.

## Accession codes

Atomic coordinates and structure factors for the Rev dimer-RRE crystal structure have been deposited in the Protein Data Bank under the accession code 4PMI.

## Acknowledgements

We are grateful to V Paavilainen, O Rosenberg, J Finer-Moore, R Stroud, L Beamer, J Fraser, C Waddling and G Meigs for assistance with crystallography and structure determination, B Adams for help with protein and RNA purification, J Fernandes for providing the Gagpol-SF2 RRE reporter construct and C Smith for help with reporter assays and viral replication assays. We thank D Booth, M Daugherty, J Fernandes and members of the Frankel lab, J Gross, L Beamer, J Fraser, Y Bai and J Doudna, for helpful discussions and comments on the manuscript. We also thank Yun-Xing Wang for providing us the coordinates for the RRE SAXS envelope. This work was supported by NIH grant P50GM082250 grant to ADF and a CHRP postdoctoral fellowship F09-SF-204 to BJ.

## Additional information

### Funding

| Funder | Grant reference number | Author |
|---|---|---|
| National Institutes of Health | P50GM082250 | Alan D Frankel |
| California HIV/AIDS Research Program | Postdoctoral Fellowship | Bhargavi Jayaraman |

The funders had no role in study design, data collection and interpretation, or the decision to submit the work for publication.

### Author contributions

BJ, Conception and design, Acquisition of data, Analysis and interpretation of data, Drafting or revising the article, Contributed unpublished essential data or reagents; DCC, Conception and design, Acquisition of data, Analysis and interpretation of data, Drafting or revising the article; CH, Acquisition of data, Analysis and interpretation of data; IR, DM, Acquisition of data, Contributed unpublished essential data or reagents; ADF, Conception and design, Analysis and interpretation of data, Drafting or revising the article

## Additional files

### Major dataset

The following dataset was generated:

| Author(s) | Year | Dataset title | Dataset ID and/or URL | Database, license, and accessibility information |
|---|---|---|---|---|
| Jayaraman B, Crosby DC, Homer C, Ribeiro I, Mavor D, Frankel AD | 2014 | Atomic coordinates and structure factors for the Rev dimer-RRE crystal structure | http://www.pdb.org/pdb/explore/explore.do?structureId=4PMI | Publicly available at RCSB Protein Data Bank. |

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
