## [Decision Letter]

Thank you for sending your work entitled “RNA-directed remodeling of the HIV–1 Rev oligomer orchestrates Rev-RRE assembly” for consideration at *eLife*. Your article has been favorably evaluated by John Kuriyan (Senior editor), Wesley Sundquist (Reviewing editor), and 2 reviewers, one of whom, Jamie Williamson, agreed to reveal his identity.

The Reviewing editor and the reviewers discussed their comments before we reached this decision, and the Reviewing editor has assembled the following comments to help you prepare a revised submission.

Jayaraman et al. present a 3.2 A resolution crystal structure of a Rev dimer bound to a fragment of RNA corresponding to the high-affinity binding site (Stem loop IIB) of the Rev-response element (RRE) plus the adjacent IIABC junction. In order to crystallize the complex, the authors used a Rev construct in which two residues were mutated at one of the hydrophobic oligomerization interfaces, promoting formation with only a Rev dimer. While dimerization of Rev occurs using the same hydrophobic interfaces as previously reported, the structure reveals that the Rev dimer binds the RNA with a crossing angle that is much more acute than was previously shown in the dimer in the absence of RNA (50 deg vs. 120-140 deg.). The authors suggest that this degree of freedom (different crossing angles) may be exploited in the formation of higher order complexes. Strong biochemical and molecular virological analyses are presented in support of the structure.

The crystal structure is the first example of a high resolution structure of a Rev dimer-RNA complex, despite considerable efforts over two decades, and is thus an important advance. Important new contributions include: 1) the discovery that the crossing angle and hydrophobic interface of the Rev dimer can change quite dramatically (∼70 degrees) in response to RNA binding, which allows the two ARM helices to bind in adjacent major grooves of the RNA, and 2) the definition of a new Rev-RNA binding mode (seen for the Rev-RNA Junction site), which may represent the second step in cooperative assembly of Rev with the RRE. The manuscript is clear and well written, and the descriptions of the structure are very lucid. The study is of high technical quality, reveals remarkable molecular flexibility in both the Rev protein configuration and its RNA interactions, and represents an important advance in our understanding of the molecular basis for HIV-1 RNA export. This and the accompanying manuscript describing the EM of a Crm1 dimer bound to Rev-RRE are the most significant advance in this important story for many years.

Major issues (which must be addressed by the authors):

This work provides the most advanced structural framework for thinking about how the Rev-RRE export complex is assembled. No additional experiments need to be performed, but the text should be modified to put the observations into better context and to address the specific issues highlighted below.

1) There are now three different arrangements of the Rev dimer that have been observed, and surely there is interesting, and perhaps confounding, structural plasticity of the Rev dimer interface. This plasticity is portrayed as being a “strategy” of the virus, but the possibility that one or more of the in vitro structures may not reflect functional complexes. The manuscript skirts around this issue, and does not really address how to build the present complex into a functional Rev RRE complex, or how all of the structures may or not be consistent with that functional complex.

2) The authors should also discuss how well the junction serves as a proxy for the junction in the viral Stem II. In particular, Stem IIA is almost certainly an A-form helix that must merge into the junction, which is only two nucleotides in the crystallized construct. It is possible that the native Stem II RNA has a different conformation, and that the plasticity of the Rev-RRE interface has allowed crystallization of a non-native conformation.

3) There is no conclusive evidence that a second molecule of Rev binds to stem IIC in the context of the larger RRE, and the data in Figure 2, panels (d) and (e) provide little support since a mutation that is absolutely required for dimerization in the Stem II40 RNA has little to no discernable effects in the context of the larger RNA. While the authors argue that, in the context of a larger RNA, this mutation's effects can be overcome is an interesting idea, there is no actual evidence. A simpler explanation is that the complex uses a different pathway for oligomerization, or even that multiple pathways exist, and if one pathway is blocked then another can be readily utilized. The authors should address this issue more directly or present data showing that the observed complex is truly a part of the oligomerization pathway.

4) Another issue that needs to be addressed more directly is how the present structure fits into the Rev hexamer structure and the previously described jellyfish model. Is it possible to accommodate the second Rev-Rev interface seen in the Rev-only crystal structure with the Rev dimer conformation seen in the present structure? Is it possible to envision a plausible hexamer structure in the context of RNA with the present dimer interface?

5) The Wang lab at NIH has proposed a specific conformation of a Rev-RRE complex based on SAXS data. How does the present structure fit into that model?

6) The final figure is problematic. It is certainly fine to speculate, but while it is formally plausible that the RRE may guide Rev into an array of dimers with multiple crossing angles upon protein-RNA contact, there is as yet no experimental basis for this and the hexamers modeled here are entirely speculative. It is equally plausible, for example, that crossing angles similarly acute to the one reported herein are ubiquitously used for RNP assembly, with larger angles used for interaction with other host-cell proteins (or other variations on the same theme). This figure should be deleted or, alternatively, replaced with a figure that explains the authors’ current general model, particularly in light of the accompanying Crm1 dimer structure.

Other issues for the authors' consideration:

1) Title. Change “Rev oligomer” to “Rev dimer”. Throughout the manuscript there should be careful usage of “Rev oligomer”, which, in at least some places here and in the accompanying manuscript, is alluding to (at least) a hexamer, and in other places to a Rev dimer. Say “dimer” when “dimer” is meant. Similarly, we suggest sticking to the term “subunit” when a single polypeptide chain is meant and not alternating with “molecule”. A dimer is also a molecule.

2) Final sentence of the Abstract: “our studies support an RRE-directed Rev oligomer assembly, where RRE utilizes the inherent plasticity of Rev interfaces and optimally positions Rev subunits to match the architecture of the Crm1 export complex”. Omit. There is nothing here about Crm1 nor about a Rev oligomer larger than a dimer.

3) The three mutations in the crystallized Rev construct should be indicated in Figure 1. The position of the oligomerization-inhibiting mutations should be indicated on the structure.

4) Figure 1—figure supplement 1 shows gel shift assays of Rev with RRE-Stem II and IIB40. It is clear the RNA below is a single species. However, the first observed shift shows a doublet band in the IIB40, and a smeared/fuzzy band in stem II. The authors should note this somehow as these are both evidence of conformational heterogeneity within their samples (the first being the presence of two stable conformations and the second a constant switching between two conformations). Further this doublet nature appears to be resolved upon the second Rev binding, indicating a resolving of this phenomenon.

5) The authors use a 240 nt RRE. They should state that the full RRE is ∼351 nt, and the smaller construct has been shown to be proficient, though not optimal, in nuclear export.

6) The Rev ARM-RNA IIB RRE interface generally agrees well with the analogous interface presented in the 1996 Battiste et al. NMR structure, but the crystal interface is described as being subtly different in detail. It seems equally likely, however, that the solution structure simply lacked precision, particularly given the state of NMR structural studies at that time. This should probably be acknowledged explicitly (or, if the authors wish to make a point of the differences, they should show that their current structure is incompatible with the earlier NMR data).

7) Figure 2C–figure supplement 1 is not useful in its current form. The intent seems to be to demonstrate more clearly than in Figure 2 a binding defects for Rev-Q51A. However, the effects are modest and it is nearly impossible for the reader to draw any conclusions from the data presented.

8) The authors should show the raw western blots that underlie the normalized data in Figure 4. The results seem clear enough, but it is important to show that the Rev mutations do not destabilize the protein significantly, and this information is lost through the normalization procedure.

9) “… enables a virus with limited coding capacity to build a large, asymmetric RNP using a small, homooligomeric protein to achieve unprecedented structural and functional complexity.” This is really just hype, given that the structure of the large asymmetric RNP is not known nor is its structural and functional complexity.

---

## [Author Response]

*Major issues (which must be addressed by the authors)*:

*1) There are now three different arrangements of the Rev dimer that have been observed, and surely there is interesting, and perhaps confounding, structural plasticity of the Rev dimer interface. This plasticity is portrayed as being a “strategy” of the virus, but the possibility that one or more of the in vitro structures may not reflect functional complexes. The manuscript skirts around this issue, and does not really address how to build the present complex into a functional Rev RRE complex, or how all of the structures may or not be consistent with that functional complex*.

While we share the reviewers’ concern that one or more of the in vitro structures may not reflect functional complexes, we note that:

All structures solved so far (including ours) have required some modification of the Rev-RRE system to enable structure determination, owing to the biochemical behavior of the system. Nevertheless, they are consistent with a large body of mutational data and functional assays and consequently are unlikely to represent irrelevant trapped states.

Our Rev-dimer/RNA configuration may be a part of the fully assembled functional complex or could represent a potential assembly intermediate. In either case, the structure illustrates the direct coupling between oligomerization and RNA binding towards cooperative assembly. The central finding of the paper, plasticity of Rev dimer interface, explains several intriguing observations in the literature (Discussion section) and connects the RNA architecture, which is a function of a continually evolving RNA sequence, to Crm1-dependent nuclear export activity and HIV replication (elaborated in the Discussion section). In fact, it will be interesting to capture Rev in other conformational states with RNA or other protein partners to observe the extent of plasticity both in this dimer interface and the second hydrophobic interface.

The present complex can be built into a full Rev/RRE complex based on the jellyfish model (Figure 5), although it is difficult to determine whether or not this is a functional complex. Our responses to other issues below are also applicable here.

*2) The authors should also discuss how well the junction serves as a proxy for the junction in the viral Stem II. In particular, Stem IIA is almost certainly an A-form helix that must merge into the junction, which is only two nucleotides in the crystallized construct. It is possible that the native Stem II RNA has a different conformation, and that the plasticity of the Rev-RRE interface has allowed crystallization of a non-native conformation*.

We recognize that stem IIA, which very likely is an A-form helix, is only two nucleotides in our construct. Several lines of evidence suggest that the junction site is a good proxy for the junction in the RRE stem II:

The position of helix disruptions adjacent to the IIB site is critical in determining Rev oligomerization and affinity (58). A disruption that matched the position of the stem II junction relative to IIB site had the highest affinity and most efficient Rev oligomerization. Furthermore, although Rev was able to dimerize on an RNA containing any helix disruption positioned next to IIB, the exact sequence of the junction site, as it is in IIB40, displayed the highest affinity and cooperativity, comparable to RRE-stem II (12) (Figure 1—figure supplement 1). Additionally, the reviewers pointed out a doublet/fuzzy band at the first observed shift in both IIB40 and stem II RNA, while the free RNAs and the Rev-dimer-bound RNAs migrate as single species.

This conformational heterogeneity in both RNAs in the Rev-monomer-bound species suggests similar binding patterns for both the RNAs. We have modified our manuscript in the first paragraph of the Results section under the title: Rev dimer-RRE structure and Figure 1—figure supplement 1 to reflect these points.

DMS probing (8) and SHAPE-seq (2) studies on Rev-RRE complexes show melting of an AU base-pair from stem IIA at the junction, with G76 potentially base-pairing with the A from the melted base-pair upon Rev binding (3). Consistent with these studies, we observe in our structure that G76 indeed base pairs with A44 (corresponding to A from the melted base pair) and not A42 (see the Results section: Diversity of RNA recognition by the Rev ARM).

As the reviewers point out, stem IIA most likely is an A-form helix that must merge into the junction. We have now included a figure (Figure 3—figure supplement 3) and corresponding text in the manuscript (in the Results section) under “Diversity of RNA recognition by the Rev ARM” to address this point. We have placed an A-form helix adjacent to the junction site, where it is likely to merge. Its position approximately determines where the remaining RRE is likely to be. If the second Rev subunit were to use its other hydrophobic interface to recruit a third Rev subunit, the ARM of the third Rev subunit would be positioned to contact the RRE, although it is possible that this interface is also plastic and could be further remodeled by the RRE.

The evidence that the junction site serves as a proxy for the RRE stem II junction is substantial, but we cannot formally rule out that its configuration differs in the context of stem IIA. Nonetheless, the basic inferences drawn from our structure, i.e. plasticity of the Rev dimer interface and the importance of the site architecture for Rev binding with few base-specific contacts, are unlikely to change.

*3) There is no conclusive evidence that a second molecule of Rev binds to stem IIC in the context of the larger RRE, and the data in*
Figure 2*, panels (d) and (e) provide little support since a mutation that is absolutely required for dimerization in the Stem II40 RNA has little to no discernable effects in the context of the larger RNA. While the authors argue that, in the context of a larger RNA, this mutation's effects can be overcome is an interesting idea, there is no actual evidence. A simpler explanation is that the complex uses a different pathway for oligomerization, or even that multiple pathways exist, and if one pathway is blocked then another can be readily utilized. The authors should address this issue more directly or present data showing that the observed complex is truly a part of the oligomerization pathway*.

These comments are related to issue (2) above, and we have made changes to the Results section under: The Rev dimer rearranges upon RNA binding, to clarify the issue and raise alternative possibilities. There are at least three explanations for the observations in Figure 2:

In the context of the full length RNA, the effect of a Q51A mutation can be overcome by other compensating interactions.

As the reviewers suggest, there could be multiple pathways for oligomerization and that if one pathway is blocked, then another can be utilized.

The observed complex is not truly part of the oligomerization pathway.

4) Another issue that needs to be addressed more directly is how the present structure fits into the Rev hexamer structure and the previously described jellyfish model. Is it possible to accommodate the second Rev-Rev interface seen in the Rev-only crystal structure with the Rev dimer conformation seen in the present structure? Is it possible to envision a plausible hexamer structure in the context of RNA with the present dimer interface?

The present structure can be used to form a Rev hexamer model, similar to the previously described jellyfish model, albeit with a different overall architecture. Figure 5, which has now been modified, depicts some plausible hexamer models that use different combinations of both the present dimer interface and the RNA-free dimer interface connected by the second Rev-Rev interface.

5) The Wang lab at NIH has proposed a specific conformation of a Rev-RRE complex based on SAXS data. How does the present structure fit into that model?

Our model from Figure 3—figure supplement 3, when placed into the SAXS envelope of the RRE, is consistent with the Wang lab’s overall map of RRE structural elements ([17]; Figure 3—figure supplement 3, and also described in the Results section under: “Diversity of RNA recognition by the Rev ARM”). Their specific model proposes that stems IIB and IA template Rev assembly on the RRE and bridge the arms of the ‘A’ shaped RRE structure. However, DMS probing (8) and SHAPE-seq experiments (2) indicate that Rev assembly nucleates at stem IIB and proceeds via stem II along the length of stem I, implying that Rev-binding at IA occurs after Rev binding to IIB and the stem II junction. Thus, our structure is consistent with the RRE chemical probing studies but does not support the specific model of an initial Rev-dimer assembling at IIB and IA. Our structure supports a more general model where the RRE topology and major groove spacing sets the arrangement of Rev oligomers, which could bridge the two arms of the RRE ‘A’ structure.

*6) The final figure is problematic. It is certainly fine to speculate, but while it is formally plausible that the RRE may guide Rev into an array of dimers with multiple crossing angles upon protein-RNA contact, there is as yet no experimental basis for this and the hexamers modeled here are entirely speculative. It is equally plausible, for example, that crossing angles similarly acute to the one reported herein are ubiquitously used for RNP assembly, with larger angles used for interaction with other host-cell proteins (or other variations on the same theme). This figure should be deleted or, alternatively, replaced with a figure that explains the authors’ current general model, particularly in light of the accompanying Crm1 dimer structure*.

We agree that the figure showed too much detail given the speculative arrangements of Rev hexamers. The intent of the figure is to highlight plausible hexamer models whose actual architecture would depend on the scaffolding RNA or host-cell proteins. To clarify, we have modified the figure as suggested to connect RNA structure and Rev plasticity with Crm1 recruitment, in light of the Crm1 dimer structure.

*Other issues for the authors' consideration*:

*1) Title. Change “Rev oligomer” to “Rev dimer”. Throughout the manuscript there should be careful usage of “Rev oligomer”, which, in at least some places here and in the accompanying manuscript, is alluding to (at least) a hexamer, and in other places to a Rev dimer. Say “dimer” when “dimer” is meant. Similarly, we suggest sticking to the term “subunit” when a single polypeptide chain is meant and not alternating with “molecule”. A dimer is also a molecule*.

We have made the suggested changes and changed the title according to the journal request. The title now is: “RNA-directed remodeling of the HIV–1 protein Rev orchestrates assembly of the Rev-Rev response element complex”.

*2) Final sentence of the Abstract: “our studies support an RRE-directed Rev oligomer assembly, where RRE utilizes the inherent plasticity of Rev interfaces and optimally positions Rev subunits to match the architecture of the Crm1 export complex”. Omit. There is nothing here about Crm1 nor about a Rev oligomer larger than a dimer*.

We have made the suggested change.

*3) The three mutations in the crystallized Rev construct should be indicated in*
Figure 1*. The position of the oligomerization-inhibiting mutations should be indicated on the structure*.

We have indicated the mutations in Figure 1 and Figure 1—figure supplement 1.

*4)*
Figure 1—figure supplement 1
*shows gel shift assays of Rev with RRE-Stem II and IIB40. It is clear the RNA below is a single species. However, the first observed shift shows a doublet band in the IIB40, and a smeared/fuzzy band in stem II. The authors should note this somehow as these are both evidence of conformational heterogeneity within their samples (the first being the presence of two stable conformations and the second a constant switching between two conformations). Further this doublet nature appears to be resolved upon the second Rev binding, indicating a resolving of this phenomenon*.

We have noted this in the legend for Figure 1—figure supplement 1.

*5) The authors use a 240 nt RRE. They should state that the full RRE is ∼351 nt, and the smaller construct has been shown to be proficient, though not optimal, in nuclear export*.

We have included this information in the Introduction section.

*6) The Rev ARM-RNA IIB RRE interface generally agrees well with the analogous interface presented in the 1996 Battiste et al. NMR structure, but the crystal interface is described as being subtly different in detail. It seems equally likely, however, that the solution structure simply lacked precision, particularly given the state of NMR structural studies at that time. This should probably be acknowledged explicitly (or, if the authors wish to make a point of the differences, they should show that their current structure is incompatible with the earlier NMR data)*.

As suggested, we indicate that the solution structure solved nearly two decades ago may have lacked precision (see Results section under: “Diversity of RNA recognition by the Rev ARM”) and does not suggest experimental differences with the current structure.

*7) Figure 2C–figure supplement 1 is not useful in its current form. The intent seems to be to demonstrate more clearly than in*
Figure 2
*a binding defects for Rev-Q51A. However, the effects are modest and it is nearly impossible for the reader to draw any conclusions from the data presented*.

The purpose of the figure is to show that although the binding defect of Q51A is modest, it is observed in two different assays: one using stoichiometric amounts of Rev and RRE (Figure 2) and the other with Rev in vast excess of the RRE (Figure 2—figure supplement 1). Hence, the observed defect is modest but real. We have clarified this in the Results section under: “The Rev dimer rearranges upon RNA binding”.

*8) The authors should show the raw western blots that underlie the normalized data in*
Figure 4*. The results seem clear enough, but it is important to show that the Rev mutations do not destabilize the protein significantly, and this information is lost through the normalization procedure*.

We presume the reviewer was referring to western blots for Figure 4, not for the replication time course of Figure 4, and these data are now included. Data were normalized both to Rev and GAPDH expression levels.

*9) Page 9. “… enables a virus with limited coding capacity to build a large, asymmetric RNP using a small, homooligomeric protein to achieve unprecedented structural and functional complexity.” This is really just hype, given that the structure of the large asymmetric RNP is not known nor is its structural and functional complexity*.

We have modified the sentence.